# Characteristic features of an active polar filament pushing a load

Prabhakar Maurya[*°], Shalabh Kumar Anand[†°] and Sunil Pratap Singh[‡]

Department of Physics, Indian Institute Of Science Education and Research,
Bhopal 462 066, Madhya Pradesh, India

⋆ mprabhakar537@gmail.com , † shalabh.kumar.anand@gmail.com , ‡ spsingh@iiserb.ac.in

## Abstract

We present the structural and dynamical behavior of an active polar filament that is pushing a load using overdamped Langevin dynamics simulations. By varying the bending rigidity and the connectivity between the filament and the load, we smoothly transition the boundary condition of the filament from pivoted to clamped. In the clamped state, the load remains strongly aligned with the filament, whereas in the pivoted state, the load is free to rotate at its attachment point. Under the pivoted boundary condition, the active polar filament buckles and exhibits various fascinating dynamical phases, including snake-like motion, rotational motion, and helical conformations. However, under the clamped boundary condition, the helical phase disappears, and the filament attains either an extended or a bent conformation. The transition from the extended state to the helical phase is characterized using a global helical order parameter in the parameter space of active force and a physical quantity associated with the boundary condition. We have obtained various power laws relating the curvature radius of the helical phase, effective diffusivity, and rotational motion of the monomers to the active force. Furthermore, we demonstrate that the filament's effective diffusivity in the helical phase exhibits a non-monotonic dependence on the active force: it initially increases linearly but decreases sharply at high active force strengths.

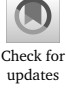

## Contents

° These authors contributed equally to the development of this work.

# 1   Introduction

Thin filamentous macromolecules play an indispensable role in living matter, notably in cell motility [1], sperm movement [2,3], bacterial locomotion [4,5], vesicle transport [6], muscle contraction [7], shape change in neurons, sensing and recasting mechanical stresses into an electrical signal in vestibular sensing cells [8], cell division [2,3,9,10], etc. These filaments experience mechanical forces, distinct from thermal fluctuations, generated by various enzymes and motor proteins while carrying out biological functions [11–13]. Such stresses can induce substantial conformational changes [2–4], leading to intriguing structural and dynamical phases, such as the beating and rotary motion of cilia and flagella, helical twisting of chromosomes during replication [14–16], bending of the stereocilia, and beating and buckling of actin bundles, etc. With these insights, considerable progress has been made in understanding the conformational dynamics of a single active filament in previous studies [2,17–30]. Microscopic buckling of filaments under active compressive or extensile stresses gives rise to intriguing macroscopic structures in dense suspensions, including polar bands, dynamic polar domains, defect formations, and highly distorted lines. These structures emerge due to variations in active stress strength [31–33], and are governed either by the Euler-buckling length scale of the filaments or by hydrodynamic instabilities.

Active polymers have been modeled predominantly using two distinct approaches. In the first case, the active noise on the monomers follows an Ornstein-Uhlenbeck process [18,27, 34–44]. In the second approach, the active process is quenched along the polymer backbone, aligned with its tangent. This category of active polymers is referred to as an active polar linear polymer (APLP) [2,45–49]. The first approach leads to a non-monotonic structural behavior, where the active forces compress the polymer followed by swelling at high force regime [34,43]. The latter approach results in a coil-to-globule-like transition with significant compression [20,45,46], particularly for flexible polymers in the limit of higher activity. A similar model has been adopted for understanding complex biological functions, including the formation of chromatin compartments, enhanced segmental dynamics of chromatin, and coherent macroscale motion of the chromatin [50–52]. A freely moving active polar filament adopts various structures, such as strongly buckled shapes, circular shapes, and helical coils [20,46,53]. While a clamped active polar filament exhibits rotational and beating motion [17,19,54–56], and its structural dynamics resemble those of a rotating flagellum.

The present work primarily assesses the variation of the boundary conditions of the load (front bead), the bending rigidity between the front bead and the rest of the filament (see Fig.1), the monomer size, and the strength of the compressive active force. These variations can lead to novel configurations, particularly the emergence of a dynamically stable helical state of active polar filaments. Helical structures are particularly intriguing, as they play an essential role in numerous biological functions. Notable examples in living matter are listed here: double-stranded DNA, actin filaments, viral capsids [57], $\alpha$-helices in protein subunits [58], helical organization of bacterial chromosomes [14–16], etc. However, it is important to note that such structures in biological systems are typically stabilized by physical interactions that differ from those described in the present study, where helical formation results from compressive active forces. Additionally, it has been established that a long straight filament

adopts a helical buckled state under compressive flow or within a viscosity gradient [59, 60]. Understanding the kinetics and thermodynamic stability of the helices is crucial, as it provides valuable insights into various biological functions.

This article presents a minimal model for the active polar filament pushing a cargo (load) in three dimensions (3D). The three-dimensional filament offers more complex emergent structures compared to a two-dimensional active filament, where filaments tend to remain trapped in spiral structures for extended periods [2, 20, 61]. The sensitivity of the structural dynamics to the boundary conditions of the front monomer and its load has already been emphasized for the case of an active polar semi-flexible polymer in 2D [62]. However, a systematic study of such a system in 3D is lacking, which could bring more fascinating phases. We comprehensively study an active filament under compressive force pushing a load using coarse-grained computer simulations. We observe the emergence of helical structures at the limit of the large active force on variation in boundary conditions. This helical state remains stable up to a certain bending rigidity associated with the load; however, for larger bending rigidities, the helical state disappears. Such structures are not feasible in the lower spatial dimensions. Specifically, the active filament in the helical state exhibits rotational motion along its axis, a behavior distinctly different from that of two-dimensional active filaments [20, 62]. Furthermore, our results show a smooth monotonic decrease in the gyration radius and the filament's end-to-end distance as the active force increases, even in the presence of the load. Importantly, this smooth monotonic decrease in the structural properties transitions into a sharp decline under variations in boundary conditions.

We determine structural transitions using various physical parameters, including the tangent-tangent correlation function, the bending energy, and the global helical order parameter ($H_4$). The correlation function exhibits an oscillatory behavior, whereas the helical order parameter displays a monotonic transition from the extended state to the helical phase. In addition, the curvature radius of the twist decreases with increasing activity. Furthermore, we analyze the dynamical behavior of the filament using mean-squared displacement (MSD) and compute the self-diffusion coefficient of the filament. Strikingly, the effective diffusion coefficient of the active filament shows non-monotonic behavior as a function of Péclet number. For large Péclet numbers, the effective diffusivity of the active filaments abruptly drops, approaching the passive limit. More importantly, the internal dynamics of the monomers reveal that in the helical phase, the motion of monomers is oscillatory, with monomers rotating along the axis of the helix. The rotational frequency of the monomer follows a power-law dependence on the Péclet number, given as $Pe^{7/4}$. This power-law variation of the oscillation frequency is described using a straightforward scaling relation as a function of the curvature radius.

The manuscript is organized as follows: The model of the active filament, with the description of the load and parameters, is provided in the Model section. The structural transition of the filament, the helical order parameter, and dynamical quantities are discussed in the Results section. Finally, we summarize our study in the Conclusion section.

## 2   Model

We model an active filament as a linear polymer of $N$ spherical Brownian monomers linearly connected through harmonic springs. These monomers interact among themselves via excluded volume interactions. The bending potential, to control the stiffness of the backbone of the filament, is also incorporated. Thus, the total energy of the filament can be written as $U = U_s + U_b + U_{LJ}$. Here, $U_s$, $U_b$, and $U_{LJ}$ are spring, bending, and excluded volume potentials,

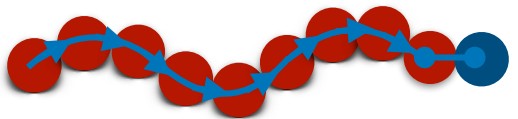

Figure 1: A schematic of an active filament with a load. Arrows indicate the direction of the active forces on the monomers. The front monomer (blue) and its connection are shown differently. The connectivity of the front monomer with filament depicts that it can rotate depending on the boundary condition.

respectively. The spring potential $U_s$ of the filament is written as

$$U_s = \frac{k_s}{2} \sum_{i=1}^{N-1} (|\boldsymbol{r}_{i+1} - \boldsymbol{r}_i| - \ell_0)^2 \,, \tag{1}$$

where $k_s$ and $\ell_0$ are the spring constant and the equilibrium bond length, respectively, and $\boldsymbol{r}_i$ is the position vector of the $i^{th}$ monomer.

The bending potential $U_b$, which provides stiffness to the backbone of the filament, is given by

$$U_b = \left[ \frac{\kappa_B}{2} \sum_{i=1}^{N-3} (\boldsymbol{R}_{i+1} - \boldsymbol{R}_i)^2 + \frac{\kappa_B^h}{2} (\boldsymbol{R}_{N-1} - \boldsymbol{R}_{N-2})^2 \right] . \tag{2}$$

Here $\boldsymbol{R}_i$ is $i^{th}$ bond vector, $\boldsymbol{R}_i = \boldsymbol{r}_{i+1} - \boldsymbol{r}_i$ and $\kappa_B$ is the bending rigidity of the filament. The stiffness of the filament is defined in terms of persistence length $l_p$ which can be expressed in terms of $\kappa_B$ as $l_p = \kappa_B \ell_0^3/(k_B T)$, with $k_B T$ being thermal energy. The boundary conditions between the head and the filament are varied by the bending rigidity $\kappa_B^h$ of the bond connecting the head (load), which may differ from the bending rigidity of the other bonds ($\kappa_B$).

The excluded volume potential, $U_{LJ}$, is implemented via the repulsive part of the Lennard-Jones potential. More specifically, for a distance $R_{ij} < 2^{1/6}\sigma$ between two monomers $i$ and $j$ is given by

$$U_{LJ} = \sum_{i>j}^{N} 4\epsilon \left[ \left( \frac{\sigma}{R_{ij}} \right)^{12} - \left( \frac{\sigma}{R_{ij}} \right)^6 + \frac{1}{4} \right] , \tag{3}$$

and $U_{LJ} = 0$ for $R_{ij} \geq 2^{1/6}\sigma$. Here, $\sigma$ and $\epsilon$ are the LJ diameter of a monomer and the LJ interaction energy between monomers, respectively. For the model's simplicity, we have disregarded the role of hydrodynamics interactions, however, it can qualitatively influence the outcome of simulations [63–65].

The dynamics of the active filament is governed by the over-damped Langevin equation

$$\gamma_i \dot{\boldsymbol{r}}_i(t) = -\nabla_i U + \boldsymbol{F}_a^i(t) + \boldsymbol{F}_t^i(t) . \tag{4}$$

Here $\gamma_i$ is the viscous drag coefficient, $\boldsymbol{F}_a^i$ is the active force, and $\boldsymbol{F}_t^i$ is the thermal noise with zero mean. Here, $\gamma_i = \gamma$ for $i \neq N$ and $\gamma_N = \gamma^h$ for the head monomer. The second moment of the thermal noise obeys the fluctuation-dissipation relation (FDT)

$$\langle \boldsymbol{F}_t^i(t) \cdot \boldsymbol{F}_t^j(t') \rangle = 6 k_B T \gamma_i \delta_{ij} \delta(t - t') . \tag{5}$$

The active force acts along the bond direction, for the $i^{th}$ bond vector is given as $f_a \boldsymbol{R}_i / |\boldsymbol{R}_i| = f_a \hat{\boldsymbol{t}}_i$, where $\hat{\boldsymbol{t}}_i$ is the $i^{th}$ unit bond vector and $f_a$ is the magnitude of active force. Thus, the active force on $i^{th}$ monomer, presented as $\boldsymbol{F}_a^i$ in Eq. 4, has contributions shared by $i^{th}$

and $(i-1)^{th}$ bonds. This can be expressed by adding contributions from both bonds, yielding $\boldsymbol{F}_a^i = \frac{1}{2}f_a(\hat{\boldsymbol{t}}_i + \hat{\boldsymbol{t}}_{i-1})$. The strength of the active force on a monomer can be expressed in terms of a dimensionless number called Péclet number, which is defined as the ratio of the active force with the thermal energy, $Pe = f_a\ell_0/(k_BT)$.

Additionally, the front monomer is considered to be different from other monomers of the filament. Therefore, the bending rigidity ($\kappa_B^h$), size ($\sigma^h$), and friction coefficient ($\gamma_t^h$) of the front monomer are taken as independent parameters. The alignment of the load with the rest of the filament can be controlled by changing the bending stiffness ($\kappa_B^h$) of the bond connected to the front-most monomer. The random alignment of the head with respect to the rest of the filament acts as a higher load to the filament; consequently, by tuning the bending ($\kappa_B^h$), we can change the strength of the load. This is presented in terms of a dimensionless parameter $\rho = \kappa_B^h/\kappa_B$ as the control parameter that changes the boundary condition through which the load is attached to the front of the filament. The boundary condition is varied by $\rho$, ranging from pivoted ($\rho = 0$) to clamped ($\rho = 1$). For $\rho = 0$, it allows the load to rotate freely from the filament axis, while for $\rho \geq 1$, the fluctuations out of the axis are suppressed. The model restores to the tangentially propelled filament for the case of $\rho = 1$, except for the front bead, which does not experience the active force.

In addition to varying the stiffness of the load, the size of the load, $\alpha = \sigma^h/\sigma$, is also varied while the bending stiffness is kept fixed as the rest of the filament, i.e., $\kappa_B^h = \kappa_B$. Here, $\alpha$ varies the diameter of the load; when doing so, we make sure that the equilibrium bond length corresponding to the load also changes as follows: $\ell_0^h = 0.5(\sigma + \sigma^h)$. The results corresponding to this aspect are discussed at the end of the manuscript.

All physical parameters in this manuscript are scaled in units of the bond length $\ell_0$, the LJ energy $\epsilon$, and the friction coefficient $\gamma$. The simulations parameters are chosen as $k_s = 1000\epsilon/\ell_0^2$, $\sigma = \ell_0$, $\epsilon/(k_BT) = 10$. The stiffness parameter $\kappa_B$, is fixed to be 1000 unless otherwise mentioned. The number of monomers, including the load, is 201. All simulations are performed in a three-dimensional space. The Euler integration method is used with a time-step in the range of $10^{-5}\tau$ to $10^{-4}\tau$ throughout to ensure the stability of the simulations. For statistical accuracy, each data set is averaged over at least ten independent runs for all the results presented in this manuscript.

# 3   Results

We first investigate the structural behavior of the tangentially driven active polar filament, focusing on the head monomer, which acts as a load, by systematically varying its connection to the filament and its size. The effect of varying the size of the head monomer is considered by incorporating different friction coefficients for the front monomer. This is expressed here as $\gamma^h = \gamma\sigma_h/\sigma = \alpha\gamma$, where $\gamma$ is the friction of the other monomers. We analyze how these changes affect the filament's conformation and dynamics. The boundary condition between the head and the filament is controlled by the bending rigidity $\kappa_B^h$, which differs from the bending rigidity of the other bonds ($\kappa_B$). We express this difference as a ratio of the bending parameters defined $\rho = \kappa_B^h/\kappa_B$. For $\rho = 0$, the bond acts as a pivot between load and filament, allowing the front monomer to rotate freely relative to the rest of the filament; we refer to this as the pivoted boundary condition. Conversely, when $\rho \geq 1$, the load is tightly coupled to the filament's orientation, called the clamped boundary condition. In the pivoted boundary condition, the head monomer's orientation is nearly random relative to the rest of the filament. As a result, a component of the active force exerted on the head monomer is directed randomly, effectively increasing drag and opposing the directed motion of the filament.

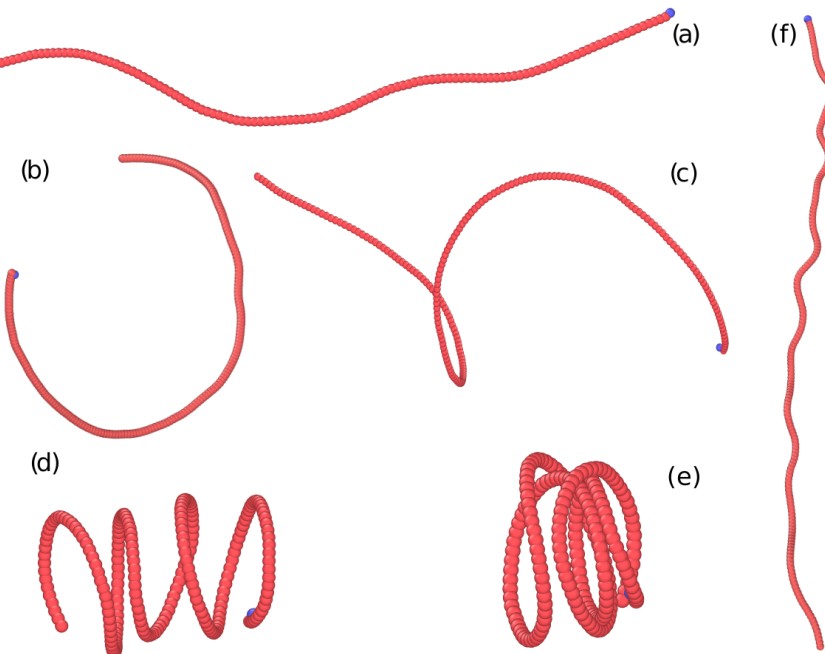

Figure 2: Various snapshots of the active polar filament for (a) $Pe = 10$, (b) $Pe = 20$, (c) $Pe = 40$, and (d) and (e) $Pe = 160$ at $\rho = 0$. f) A conformation of the filament for $\rho = 1$ at $Pe = 400$. The front monomer is notably presented in blue color.

## 3.1 Helical transition

A long, straight filament under a compressive active force tends to buckle, resulting in correlated or uncorrelated buckling of the filament [46]. The filament transitions through various structures, from rod-like to helical conformations, as illustrated in Fig. 2. We calculate the helical order parameters to quantify structural transitions. For this we follow the definition of Ref. [66–68] to compute the global ($H_4$) and local ($H_2$) helical order parameters

$$H_4 = \left\langle \left( \frac{1}{N-2} \sum_{i=1}^{N-2} \hat{\mathbf{u}}_i \right) \cdot \left( \frac{1}{N-2} \sum_{i=1}^{N-1} \hat{\mathbf{u}}_i \right) \right\rangle,$$

$$H_2 = \frac{1}{N-3} \left\langle \sum_{i=1}^{N-3} \hat{\mathbf{u}}_i \cdot \hat{\mathbf{u}}_{i+1} \right\rangle. \tag{6}$$

Here, $\mathbf{u}_i$ is a vector product of two successive bond vectors, i.e., $\mathbf{u}_i = (\mathbf{r}_{i+1} - \mathbf{r}_i) \times (\mathbf{r}_{i+2} - \mathbf{r}_{i+1})$ and $\hat{\mathbf{u}}_i = \mathbf{u}_i/|\mathbf{u}_i|$. As per the definition, $H_4$ measures the global twist of the filament, whereas $H_2$ characterizes the local twist of the filament.

The computed values of $H_4$ are presented in Fig. 3(a) for various $\rho$ as a function of Péclet number (Pe). For $\rho \leq 1$, $H_4 \approx 0$ suggests that the filament does not exhibit a global twist. If the bending rigidity of the filament with the load is equal to or greater than the rest of the filament, the global buckling does not occur. Furthermore, as $\rho$ decreases, $H_4$ increases sharply. As shown in the plot, $H_4$ suddenly rises from zero to a large value as Pe increases. In the limit of large Pe, $H_4$ almost saturates to a common value for all $\rho < 0.3$. The appearance of non-zero values in $H_4$ indicates a structural transition for each $\rho$. Interestingly, the filament exhibits local bending for $\rho \geq 1$ despite $H_4 \approx 0$. This local bending can be measured in terms of $H_2$, which increases with increasing Pe, as shown in Fig. 3(b). However, despite local bending, the filament does not attain the helical shape, and the global order parameter remains $H_4 \approx 0$. Moreover, for $\rho < 1$, $H_2$ also increases sharply and reaches a common plateau value for all $\rho$.

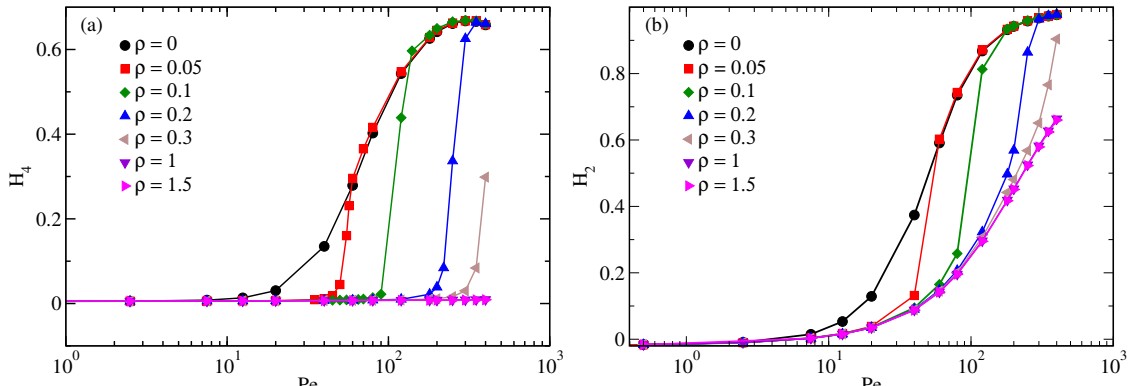

Figure 3: a) Average global helical order parameter $H_4$ (Eq. 6) as a function of $Pe$ for different $\rho$. b) The variation of the average local order parameter $H_2$ (Eq. 6) as a function of $Pe$ for various $\rho$ as indicated in the legend.

The key difference between $H_4$ and $H_2$ is that in the extended phase, $H_4 \approx 0$ whereas $H_2$ is non-zero and grows with $Pe$ for all $\rho$, see Fig. 3(b).

## 3.2 Bending energy

The helical buckling of the filament is achieved under compressive force, resulting in higher bending energy than an extended filament. The structural transition of the filament can also be identified by its bending energy. To illustrate this, we present the average bending energy $U_b$ of the filament in Fig. 4 as a function of $Pe$ for various values of $\rho$.

The bending energy remains unchanged for $Pe < 10$, for the covered range of $\rho$. Beyond $Pe > 10$, a systematic monotonic increase in the bending energy appears, as Fig. 4 displays for $\rho = 0$. For $\rho > 0$ values, the plateau range extends beyond $Pe > 10$. Importantly, a sharp transition in bending energy is observed at a critical Péclet number for $\rho > 0$. At this juncture, the bending energy curves merge in that of $\rho = 0$. A sharp upsurge in the bending energy for $\rho < 1$ at the critical Péclet number indicates that the filament undergoes the structural transition from the extended state to a buckle state, in this case, a helical state, as Fig. 3-(a) also displays the spontaneous emergence of non-zero values of $H_4$. Importantly, the transition from the extended state to the helical state for $\rho = 0$ is gradual rather than sudden, as in the case of $\rho > 0$. The bending energy in the helical phase grows with a power law, given as $U_b \sim Pe^{4/3}$, a dashed line illustrates this variation in the limit of $Pe > 10$. We would like to mention that the scaling of bending energy with activity differs from that in the context of 2D filament pushing a load, as the bending energy scales linearly with activity in 2D [62]. Although the scaling relation differs in 2D and 3D, in the high activity regime, the energies for smaller loads (high $\rho$) merge with those of the high-load curves in both cases.

The transition point, marked by a sharp bending energy increase, shifts toward higher $Pe$ as $\rho$ increases. Notably, we observe two universal bending energy curves, one corresponding to the helical shape and the other to the extended-state or folded conformation. A change in the bending rigidity of the load oversees the transition from a lower bending energy state to a higher bending energy state. This reveals that, despite having a higher bending energy, the helical phase remains a stable state, and these conformations are stabilized by the compensation of compressive force with viscous drag.

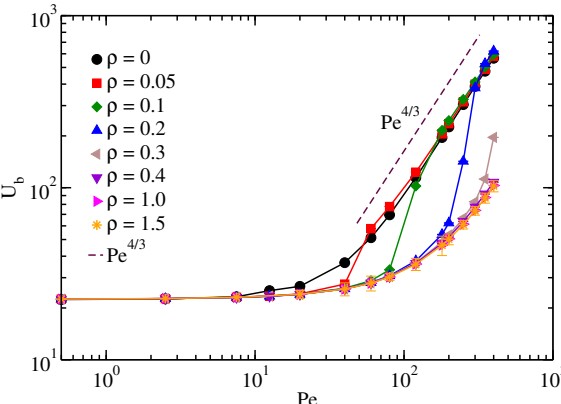

Figure 4: The variation of the bending energy ($U_b$) as a function of $Pe$ for various $\rho$ as indicated in the legend. The dashed line illustrates the power law behavior of the bending energy ($U_b \sim Pe^{4/3}$) with an exponent 4/3 in the helical phase.

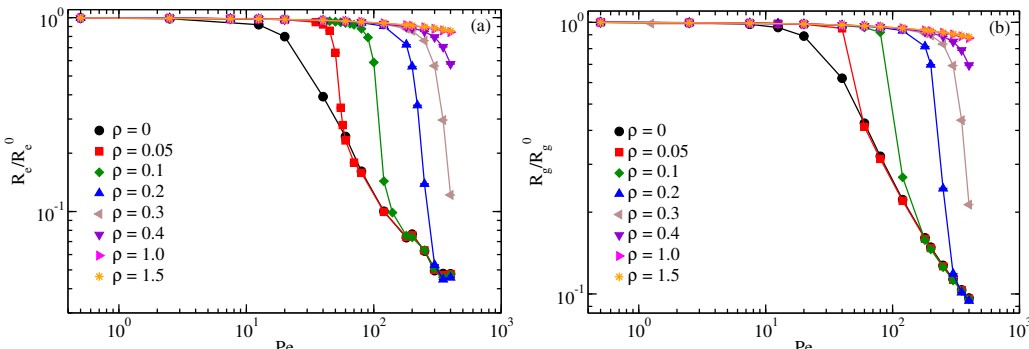

Figure 5: a) The variation of the average end-to-end distance $R_e/R_e^0$ as a function of Péclet number $Pe$. b) The average radius of gyration $R_g/R_g^0$ of the filament as a function of Péclet number $Pe$ for various $\rho$ as displayed in the plot.

## 3.3 Structural properties

We illustrate the change in the structure of active polar filament using average end-to-end distance ($R_e$) and radius of gyration ($R_g$). These are expressed as,

$$R_e = \langle \sqrt{(\boldsymbol{r}_1 - \boldsymbol{r}_N)^2} \rangle,$$

$$R_g = \left\langle \sqrt{\frac{1}{N} \sum_{i=1}^{N} (\boldsymbol{r}_i - \boldsymbol{r}_{cm})^2} \right\rangle, \tag{7}$$

where $\boldsymbol{r}_{cm}$ is the center of mass of the filament, and angular brackets represent the ensemble average. Figure 5 displays variation of the normalized values of $R_e/R_e^0$ and $R_g/R_g^0$ in the parameter space of $Pe$ and $\rho$.

For the pivot boundary condition $\rho = 0$ and small activity $Pe < 10$, as expected, conformations are almost unperturbed; therefore, $R_e/R_e^0$ and $R_g/R_g^0$ remain unchanged. However, beyond a critical $Pe > 10$, $R_e/R_e^0$ and $R_g/R_g^0$ monotonically decrease, see Fig. 5 (a) and (b), respectively. The filament is substantially compressed for larger values of $Pe$, as $R_g$ and $R_e$ show a significant reduction from its equilibrium value in the presented $Pe$ range.

Furthermore, an increase in $\rho$ results in a crossover from a plateau to a compression of $R_e$ and $R_g$ for larger values of $Pe$. Importantly, the compression of the filament appears abrupt,

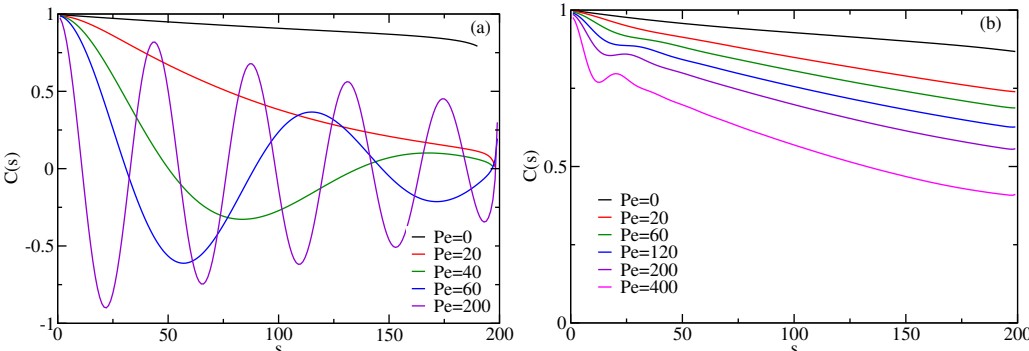

Figure 6: The tangent-tangent correlation function $C(s)$ of the active polar filament for various $Pe$ strengths at a) $\rho = 0$ and b) $\rho = 1$, here, $s = 0$ and $s = N$ correspond to tail and head monomers of the filament, respectively.

as also seen in the sharp variation of the bending energy (see Fig. 4) and the helical order parameter (see Fig. 3). For sufficiently large $Pe$, all curves merge with the behavior of $\rho = 0$. In the limit of the clamped boundary condition $\rho \geq 1$, where the head/load is rigidly connected to the filament, the structure of the active filament is nearly unchanged in the simulation window presented, indicating that the filament remains in the extended state. Therefore, no structural transition is observed here. In the limit $\rho > 1$, the active force can lead to bending of the filament; thus, compression is observed [2, 46].

## 3.4 Bond correlation

The monotonic compaction of $R_e$ and $R_g$ and the helical order parameter indicate that the filament acquires a folded/buckled conformation. The snapshots and supporting media files (see Movie 1, 2, and 3) [69] corroborate our claim of the structural transitions.

We look into spatial correlation along the contour to compute the buckling curvature of the filament. For this, tangent-tangent correlation is a useful metric. The correlation at separation of arc length $s = |i - j|\ell_0$ can be computed as $C(s) = <\hat{t}_i.\hat{t}_j>$. The correlation is computed from the rear end of the filament towards the front load, i.e., from $i = 1, \ldots, N$, and N is the load/head monomer. In equilibrium, the tangent correlation exponentially diminishes, providing the filament's persistence length $l_p$, as correlation obeys exponential behavior, $C(s) \sim \exp(-s/l_p)$. The exponential behavior is nicely captured in Fig. 6 for $Pe = 0$, particularly below the critical $Pe$ for all $\rho$ [43].

The tangential correlation diminishes substantially upon augment of Péclet numbers, indicating the lateral fluctuations along the contour are increased [46]. Interestingly, for $Pe \geq 40$, an exponential to damped oscillatory correlation emerges. The oscillation in $C(s)$ becomes more prominent and continues along its backbone for large $Pe$. The transition from exponential behavior to sinusoidal oscillations is a signature of the underlying helical folding of the filament [46, 70].

For $\rho \geq 1$, Fig. 6-(b) illustrates that the sinusoidal oscillation in the correlation is absent, even in the limit of very large Péclet numbers $Pe > 100$. Rather, a two-step decay appears in the correlation function with a kink, which becomes more prominent for larger $Pe$. This kink appears due to the buckling of the filament under a large compressive force. The front load resists forward movement, causing the filament to buckle from tail to head. This deformation is spread out along the backbone. Figure 2-e illustrates a conformation of buckled filament. Similar behavior has also been reported for a flexible active polar polymer, where the tangent-tangent correlation exhibits a negative correlation, indicating local compression of the polymer [36, 48, 49].

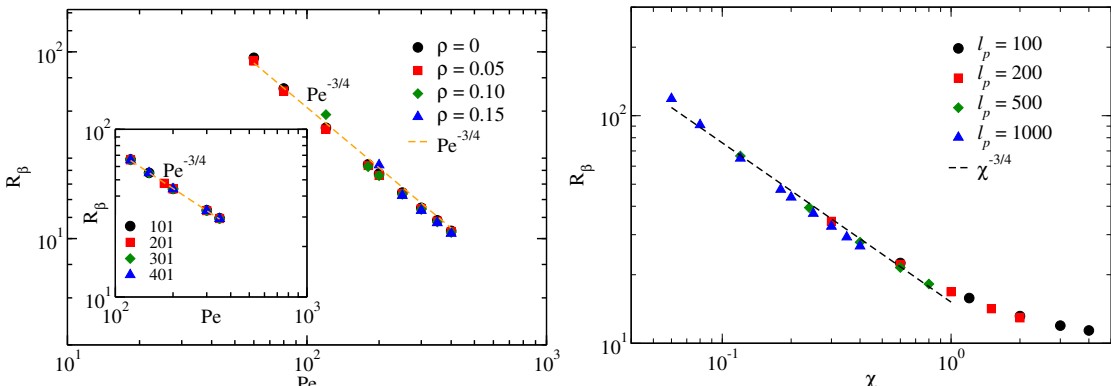

Figure 7: a) The radius of curvature ($R_\beta$) of the filament, obtained from Eq. 8, as a function of the Péclet number $Pe$ for various values of $\rho$. The dashed line shows the power law behavior of the curvature radius $R_\beta \sim Pe^{-\beta}$ with an exponent $\beta = 3/4$. Inset displays the curvature radius $R_\beta$ as a function of $Pe$ for various chain lengths at $\rho = 0$. b) Radius of curvature ($R_\beta$) as a function of the flexure number, defined as $\chi = (Pe\ell_0)/l_p$, for various bending stiffness of the filament at $\rho = 0$.

The oscillatory behavior of the tangent-tangent correlation function can provide an approximate measure of the curvature radius of the filament. To estimate this, we examine the tangent-tangent correlations by fitting a damped oscillatory function,

$$C(s) = a_\beta \exp(-s/l_p)\cos(2\pi s/R_\beta). \tag{8}$$

Here, $a_\beta$ is a constant, $l_p$ is the persistence length of the filament, and $R_\beta$ is the characteristic length scale associated with the radius of curvature of the filament in the helical phase. Figure 7 displays the curvature-radius ($R_\beta$) obtained by fitting Eq. 8 to the correlation function $C(s)$. The curvature radius decreases with activity; more importantly, a universal curve as a function of $Pe$ is obtained of all $\rho$. The obtained curvature radius shows a power law behavior $R_\beta \sim Pe^{-\beta}$ with an exponent $\beta \approx 3/4$. This characteristic feature of the curvature radius is also consistent for longer filament lengths, as illustrated in the inset of Fig. 7-a. The curvature radius is independent of the polymer length; a similar feature has also been reported for the passive filament under compressive flow [59].

Furthermore, we compute $R_\beta$ for different bending stiffnesses (persistence lengths) of the filament. Strikingly, curves for various $l_p$ overlap and demonstrate the universal behavior as a function of flexure number ($\chi$), which is defined as $\chi = \ell_0 Pe/l_p$. Notably, the curvature radius $R_\beta$ retains the same scaling behavior for all the bending stiffness with an exponent $\beta \approx 3/4$. A crossover from the power law regime to a plateau value is observed for larger values of $\chi > 1$.

## 3.5 Dynamics

The dynamics of an active filament is characterized by the mean-squared displacement (MSD) of the center of mass. Insight into the internal dynamics can be gained by examining monomers' MSD. For this first, we compute the MSD of the center of mass of the filament from the expression $<\Delta r_{cm}^2(t)> = <[r_{cm}(t) - r_{cm}(0)]^2>$, here angular bracket stands for the ensemble average. The MSD of an active filament exhibits a ballistic regime at short time scale, $<\Delta r_{cm}^2(t)> \sim t^2$, and a diffusive regime at long time scales, $<\Delta r_{cm}^2(t)> = 6D_p t$.

Figure 8 displays the effective diffusivity ($D_p$) of the filament scaled with the diffusivity of the passive filament ($D_p^0$) estimated from the diffusive regime of the MSD. The effective diffusion coefficient, as expected, grows linearly with $Pe$, in the extended state of the filament for $\rho \geq 1$. This linear behavior of the effective diffusivity has been addressed for the case

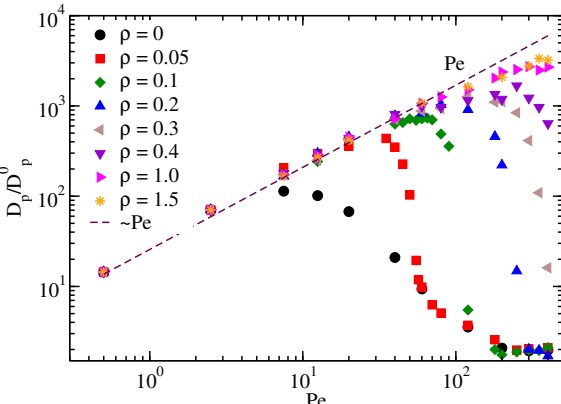

Figure 8: The plot shows scaled effective diffusion coefficient ($D_p/D_p^0$) as a function of $Pe$ for various $\rho$. The dashed line illustrates the linear behavior of scaled effective diffusivity $D_p/D_p^0$ as a function of $Pe$.

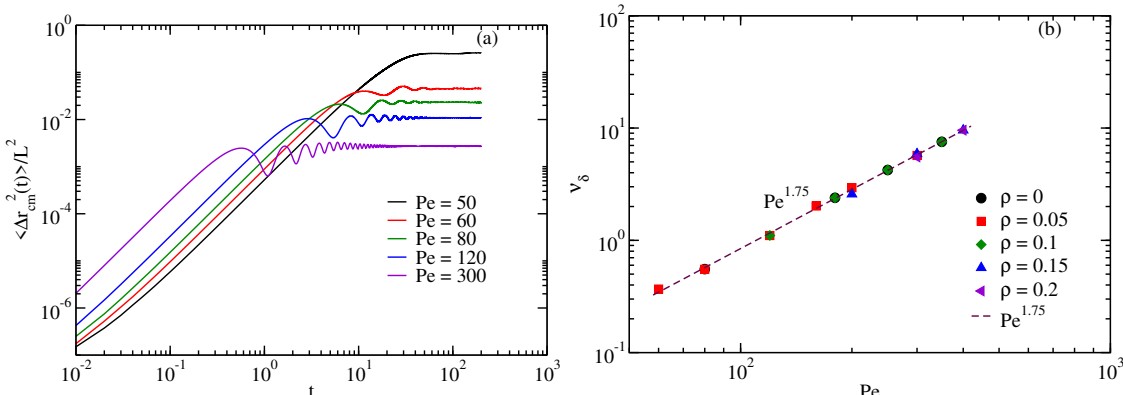

Figure 9: a) Mean-squared-displacement (MSD) of the end monomer with respect to the center of mass of the filament scaled with the contour length for various $Pe$ at a given $\rho = 0.05$. b) The rotation frequency is computed from oscillations of the MSD of the monomers for various $\rho$. The dashed line represents the power-law variation of the frequency $\nu_\delta \sim Pe^{7/4}$.

of active polar linear polymer (APLP) in the simulations and theory [45–47]. This effective diffusivity can be expressed as $D_p = D_p^0(1 + a_\rho Pe)$, where $a_\rho$ is a constant.

For $\rho < 1$, the effective diffusivity also grows linearly, superimposing with the other curves. However, it deviates from the linear regime, sharply descending beyond a critical Pe. Furthermore, in the limit of large $Pe > 200$ and $\rho < 0.3$, effective diffusivity reaches a plateau value that is very close to the passive limit, as Fig. 8-(b) depicts. The decrease in $D_p$ is due to the helical buckling of the filament, where the active force acts along the curved conformations, suppressing the large-scale directed motion of the filament. Consequently, the effective diffusivity is substantially suppressed in comparison to those for the elongated conformations. Thus, the helical shape diffuses more slowly than the extended state filament. Due to the crossover regime in effective diffusivity, a master curve over the full range of $Pe$ may not be recovered, unlike in previous studies where the Pélcet number was redefined using the radius of gyration [71].

We analyze the internal dynamics of the filament using the MSD of monomers in the considered parameter regime. Figure 9 displays the MSD of the front monomer in the center-of-mass reference frame. As expected, the MSD of the monomer grows superdiffusively before

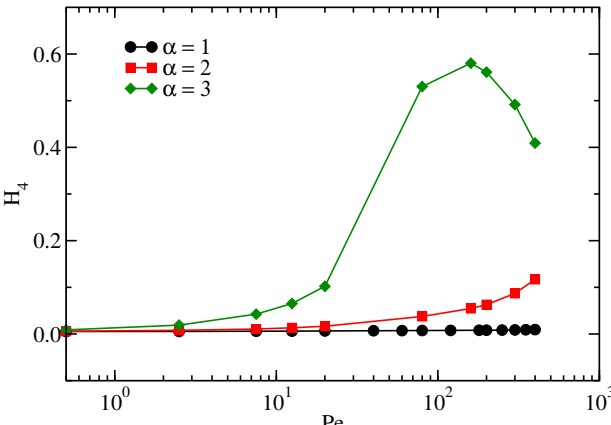

Figure 10: The global helical order parameter $H_4$ plotted as a function of $Pe$ at $\rho = 1$ for various load sizes $\alpha = 1, 2$, and 3.

approaching a plateau for all $Pe$. The plateau regime of the MSD indicates that the monomers can diffuse up to a maximum of the filament length.

Markedly, in the intermediate $Pe$ regime, MSD exhibits an oscillatory behavior just before approaching the saturation limit. This intriguing oscillatory component of the MSD appears in the helical phase, indicating that the monomers undergo rotational motion around the axis of the helical conformations. Thereby, the filament also rotates clockwise/anticlockwise with a similar frequency. The rotational motion becomes more prominent for larger Péclet numbers with larger values of oscillation frequency. The rotational motion of the monomer is illustrated in the supporting movie 3 [69].

Figure 9-(b) presents the rotational frequency of monomers ($\nu_\delta$) in the helical phase. The frequency $\nu_\delta$ increases with the Péclet number following a power law, given as $\nu_\delta \sim Pe^\delta$, with an exponent $\delta \approx 7/4$. More importantly, $\nu_\delta$ is independent of $\rho$ likewise $R_\beta$. The oscillation frequency can be described by the following scaling arguments: the speed $v_m$ of a monomer in the helical phase can be expressed as $v_m \approx R_\beta \nu_\delta$, where $R_\beta$ is the curvature radius. Employing the scaling relation of $R_\beta$ obtained in Fig. 7 and assuming that the speed of the monomer grows linear in Péclet number $v_m \sim Pe$, we can express $\nu_\delta \approx Pe/R_\beta$. Using the scaling relation obtained for the curvature radius, we obtain $\nu_\delta \approx Pe^{7/4}$, which reveals a similar exponent as obtained in simulations, see Fig.9-(b).

### 3.6 Size of the load

Now, we vary the load size while keeping $\kappa_B^h = \kappa_B$, which maintains $\rho = 1$ in our simulations. Figure 10 compares the helical order parameter $H_4$ for three different loads. For $\sigma^h/\sigma = 1$, we observe no helix formation, as discussed in the previous sections of the manuscript. Thus, the helical order parameter is nearly zero ($H_4 \approx 0$). For the load size of $\sigma^h/\sigma = 2$ and 3, we observe an increase in the values of $H_4$ as we have seen for the smaller values of $\rho < 1$. This suggests that the filament transitions to a helical structure as the load increases. Our analysis of the diffusion coefficient and radius of gyration further supports this observation, showing trends comparable to those previously discussed. Thus, the behavior of the active polar filament with a larger load is qualitatively similar to that of a filament with a pivoted boundary condition and favors the helical structure.

# 4  Summary and conclusion

We have presented the conformational and dynamical behaviors of the active polar filament pushing against a load. Different emergent phases of the filament are attained by systematically varying the load size or tuning the filament's connectivity to the load. Our findings reveal intriguing structural transitions: under sufficiently high active forces, the filament adopts helical conformations despite the absence of torsional rigidity. These helical structures emerge as stable conformations, even though they have a higher bending energy than the extended state. The helical phase remains stable over a wide range of Péclet numbers and bending parameters ($\rho$) of the load. The transition from extended to helical conformations is marked by non-zero values of the global helical order parameter ($H_4$) and a sharp increase in the bending energy of the filament. Compressive load induces helical buckling due to the combined effects of active force and viscous drag [46, 70].

In this study, we have modeled the filament load by changing the boundary condition at the microscopic level. A slight variation in the boundary condition leads to a large-scale macroscopic influence on the structure and dynamics of the filament. A smaller bending rigidity of the head monomer allows more rotational freedom. Hence, the active force on the head is not aligned with the filament's axis, effectively acting as a load on the filament against the compressive force from the rest of the filament. The effective higher load on the filament occurs due to the filament's random alignment, which causes the filament's helical buckling. As the bending stiffness at the front is raised, the fluctuations in the bond orientations of the first monomer are substantially suppressed. Consequently, the helical structure disappears in this parameter regime. For $\rho \geq 1$, a higher bending stiffness at the front bead aligns it with the filament, resulting in no significant structural changes.

Furthermore, we determined that the curvature radius of the filament in the helical phase decreases with increasing compressive active force. The behavior of curvature radius follows a power law, $R_\beta \sim Pe^{-\beta}$, where $\beta \approx 3/4$. Additionally, the filament displays intriguing dynamical behavior. In particular, the effective diffusion coefficient of the active filament shows a non-monotonic dependence on the Péclet number. For very large Péclet numbers, the effective diffusivity of the active filament abruptly drops to near-passive values despite a strong active force. A closer analysis of the internal dynamics revealed that monomers undergo oscillatory motion around the helical axis, leading to a sharp drop in the effective diffusivity. The corresponding rotational frequency follows a power-law on Péclet number, given as $\nu_\delta \sim Pe^{7/4}$.

Additionally, we have also examined the effect of varying the load size while keeping the boundary condition fixed. In this scenario, the filament also displays the transition from the extended state to the helical phase due to higher viscous drag. Thus, a larger drag on the load, whether caused by a change in boundary condition or a larger load size, results in large-scale macroscopic structural and dynamical transitions. At this point, it would be imperative to discuss our results in the context of existing literature on a filament pushing a load in two dimensions [62]. In 2D, a filament exhibited structural transitions, displaying elongated, beating, rotational, and mixed phases of circular and beating motions. A very obvious similarity between 2D and 3D occurs at low activity regimes, where the structural changes are mainly driven by thermal fluctuations. In this regime, the filament remains in an elongated phase both in 2D and 3D. However, conformations adapted by the filament in the intermediate and high activity regimes show the differences. In contrast to a 2D filament, our model does not incorporate essential interactions such as torsional rigidity, consequently beating dynamics is absent [19]. While the circling or rotating regime observed in 2D at intermediate Péclet numbers, this transitions into rotatory motion in 3D. A helical buckling becomes more favorable in 3D, making planar circling less prominent. Nevertheless, traces of circular motion can still be observed in 3D (see Fig. 2-b). A very significant difference appears at very high activity,

where our three-dimensional simulations reveal the emergence of a helical phase at high activity strength enabled by the extra degree of freedom. In both dimensions, conformational transitions result from buckling instabilities caused by compressive loads. Moreover, the scaling behavior of bending energy differs. The bending energy in the beating phase increases linearly with active force in 2D [62], whereas it follows a power-law growth with an exponent of 4/3 in the buckled regime in the 3D case discussed in this manuscript. Similarly, the power-law exponent of the rotational frequency on active force is also different, with 4/3 in 2D and 7/4 in 3D. Additionally, a key distinction between 2D and 3D cases also lies in the filament's response to tangential activity even in the absence of a load. In 2D, the filament (at small or intermediate bending rigidity) spontaneously forms a spiral shape even without any load [20]. However, in 3D, the formation of a helical shape appears in the presence of a load in the regime of high activity.

In summary, our results demonstrate how connectivity, rigidity, and load size influence the structural and dynamic behavior of the filament. These findings can offer insight into the mechanical behavior of natural microswimmers and contribute to the design of artificial swimmers capable of pushing loads [2, 3, 72]. Furthermore, solvent-mediated hydrodynamic interactions and dynamics on curved surfaces can present opportunities to address new open questions in studying such systems. In particular, the anisotropic drag of elongated objects can significantly impact the stability of various structures and dynamic behavior of the filament, especially when subjected to compressive forces [63].

## Acknowledgments

The computational facilities at IISER Bhopal are highly acknowledged for providing computational time.

**Funding information** SPS and PM acknowledge financial support from the DST-SERB Grant No. CRG/2020/000661.

**Code availability** All the simulations are performed using homemade Fortran codes. The code required to perform the simulations is available in the repository [69].

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
