# Peer review of "Characteristic features of an active polar filament pushing a load"

_SciPost Physics, doi:SciPost Phys. 18, 189 (2025)_

## Round 2 · Referee Report · Anonymous (Referee 2) · 2025-3-6

Report
The authors have positively addressed my comments. I would recommend for publication once the following minor points have been addressed:
1- at pag.3 the authors say "This article presents a minimal model for the active polar filament in three dimensions (3D) with the active force applied along its backbone" which is not true. This article is presenting a minimal model about the polymer pushing on a cargo. Please rephrase
2- the authors say "The sensitivity of the structural dynamics to the boundary conditions of the front monomer and its load has already been emphasized for the case of polar semi-flexible and flexible polymers" hence they should state why and how the present work is going beyond.
3- the authors say "Our results show a monotonic decrease in both the radius of gyration and the end-to-end distance of the filament as the active force increases." These results have already been reported and discusses. The authors should focus on the novelty of their work and not give the reader the feeling that they are redoing what is already in the literature.
4- in Fig.5 I suggest to normalize R_e and R_g by their equilibrioum values
5- in Fig.9 I sugget to rescale the MSD by the countour lenght (as indeed suggested also in the main text)
Recommendation
Ask for minor revision
Comment: at pag.3 the authors say "This article presents a minimal model for the active polar filament in three dimensions (3D) with the active force applied along its backbone" which is not true. This article is presenting a minimal model about the polymer pushing on a cargo. Please rephrase
Response: We thank the reviewer for this suggestion. We have rephrased this sentence in the revised manuscript.
Comment: the authors say "The sensitivity of the structural dynamics to the boundary conditions of the front monomer and its load has already been emphasized for the case of polar semi-flexible and flexible polymers" hence they should state why and how the present work is going beyond.
Response: We thank the reviewer for pointing out this. The boundary condition is important in defining the behavior of an active filament. For example, an active filament with follower forces exhibits beating and rotational motion. We have rephrased this sentence in the revised version and stated the importance of our work specifically in the Introduction section.
Comment: the authors say "Our results show a monotonic decrease in both the radius of gyration and the end-to-end distance of the filament as the active force increases." These results have already been reported and discusses. The authors should focus on the novelty of their work and not give the reader the feeling that they are redoing what is already in the literature.
Response: Indeed, the effect of active forces on a filament's structure is already reported. However, we report here the variation of $R_e$ or $R_g$ with $Pe$ for varying the parameter $\rho$, which is beyond the scope of the previous work. To avoid confusion, we have rephrased the sentence more clearly and highlighted our contributions in the revised manuscript.
Comment : in Fig.5 I suggest to normalize $R_e$ and $R_g$ by their equilibrium values
Response: We thank the reviewer for this suggestion, we updated the plot in the revised manuscript (see the figure attached).
Comment: in Fig.9 I suggest to rescale the MSD by the countour lenght (as indeed suggested also in the main text)
Response: We have rescaled Fig.9a as per suggestion (see the figure attached).

Anonymous on 2025-02-12 [id 5216]
Warnings issued while processing user-supplied markup:
Add "#coerce:reST" or "#coerce:plain" as the first line of your text to force reStructuredText or no markup.
You may also contact the helpdesk if the formatting is incorrect and you are unable to edit your text.
Thank you! This is a big step in the right direction. However there are problems with the code.
Running the code:
It seems that the code submitted into zenodo under https://zenodo.org/records/14836854 contains syntax errors.
Compiling with:
gfortran active_polymer.f90gives:
active_polymer.f90:15:36:15 | real(8) :: sigma(n), fric(n),b(n),rc, rc2| 1Error: Symbol ‘rc’ at (1) already has basic type of REALLooking at the code itself:
1.
I was unable to compile the code, but it appears to me that it doesn't support changing number of monomers in the form included in the repository. Looking at lines 183 and onward looks like the number of beads is hard-coded to be 50.
do ix = 50 ,xboxl-1,3do iy = 50 ,n+49do iz = 20 ,n +19i = i+1if ( i .le. n) thenx(i) = ixy(i) = iyz(i) = izendifend doend doend do2.
The code computing Lenard-Jones forces appears to be
r2=rx**2+ry**2+rz**2r2i = ss2 / r2r6i = r2i**3wij = r6i * ( r6i - 0.5 )fxij = rx * fijWhich for particles separated only along the $x$ axis by distance $r$ would give force
3.
In the code provided there are parameters such as
yboxl=n+100temp=0.1factor=1.0It seems this code works in periodic boundary conditions with some box size (there are code lines such as
rx=rx-xboxl*nint(rx/xboxl))- this is is not mentioned in the submitted manuscript. I would guess variabletemphas something to do with temperature? - this is not explained in the manuscript What is the role offactorin this code?More generally:
Lack of documentation and uploading the source file as a single file nested inside a zip makes improving this software harder than it has to be.
Please, take a look at any of the repositories from SciPost Physics Codebases: https://scipost.org/journals/publications?journal=SciPost%20Physics%20Codebases they contain at minimum: 1. detailed README with explanations of code parameters 2. detailed installation instructions
The quality of any numerical manuscript depends heavily on the quality of the accompanying software. Given the care evident in the presentation and formatting of the manuscript text, I am confident that the authors are capable of producing software code that is equally clear and well-structured. Please also include the codes used for post-processing the trajectories (for example to compute $H_2$ and $H_4$ statistics).
Anonymous on 2025-02-21 [id 5240]
(in reply to Anonymous Comment on 2025-02-12 [id 5216])Comment: It seems that the code submitted into zenodo under https://zenodo.org/records/14836854 contains syntax errors. Compiling with: gfortran active_polymer.f90 gives: active_polymer.f90: 15:36: 15∣real(8)::σ(n),ic(n),b(n),rc,rc2 ∣1 Error: Symbol ‘rc’ at (1) already has basic type of REAL
Response: We thank the reviewer for pointing out this. We have corrected the error. There was an uncommented line at line number 12, which we have removed now. Now it runs fine. We have created a new version in the repository with the updated code, initial configuration, and snapshot.
Comment: I was unable to compile the code, but it appears to me that it doesn't support changing number of monomers in the form included in the repository. Looking at lines 183 and onward looks like the number of beads is hard-coded to be 50.
Response: These lines had nothing to do with the changing the number of monomers. This was a part of initialization of the monomers of the polymer. To avoid the confusion, we have removed this part from the code. Now, we have included an input file "input_config.dat" and read the file to get the positions of the monomers. One can choose any initial configuration of a linear polymer chain and it should be fine.
Comment: The code computing Lenard-Jones forces appears to be r2=rx2+ry2+rz2 r2i=(ss2)/(r2) r6i=r2i3 wij = r6i * ( r6i - 0.5 ) fxij = rx * fij
Which for particles separated only along the x axis by distance r would give force f=r((1(r12)) - (1(2r6)) ). but normally Lenard-Jones force is f=(1/r)((1(r12)) - (1(2r6)) ).
Response: We believe the reviewer has misunderstood the code. The force along the x-axis is (rx/r2)((1(r12)) - (1(2r6)) ). This is written in the code as fij * rx, with fij = wij/r2 = rx/r2(r6i * r6i-0.5 * r6i). If we consider particles separated only along the x axis by distance r, the force we get would be f=(1/r)((1(r12)) - (1(2r6)) ).
Comment: In the code provided there are parameters such as yboxl = n+100; temp=0.1; factor=1.0 It seems this code works in periodic boundary conditions with some box size. I would guess variable temp has something to do with temperature? - this is not explained in the manuscript What is the role of factor in this code?
Response: We have made a comment about the factor in the code, which is a ratio of load diameter to the monomer diameter. "factor=1" means the diameter of all the monomers are same. The parameter "temp" is temperature, which is mentioned in the manuscript in units of the LJ energy kBT/epsilon. All the simulations are performed without hydrodynamic interactions; therefore, periodic boundary conditions are not relevant here. The uploaded code had this feature. However, to avoid such confusion, this has been removed.
Comment: Please, take a look at any of the repositories from SciPost Physics Codebases: https://scipost.org/journals/publications?journal=SciPost%20Physics%20Codebases they contain at minimum: 1. detailed README with explanations of code parameters 2. detailed installation instructions
The quality of any numerical manuscript depends heavily on the quality of the accompanying software. Given the care evident in the presentation and formatting of the manuscript text, I am confident that the authors are capable of producing software code that is equally clear and well-structured. Please also include the codes used for post-processing the trajectories (for example to compute H2 and H4 statistics).
Response: We have updated a new version in zenodo, which also has a readme file with the details and instructions to run the code, an output file in xyz form to visualize the trajectory and a snapshot from the simulation of the code. The calculation of H2 and H4 are routine analysis, we have defined the quantities properly and also cited a couple of articles where it has been done.
Anonymous on 2025-03-20 [id 5301]
(in reply to Anonymous Comment on 2025-02-21 [id 5240])Thank you for the updates submitted to Zenodo.
I can confirm that the code can be compiled and runs, and that the authors provided a minimal README file which lists parameters and their interpretations.
I understand that computation of H2 and H4 is relatively straightforward, but not including it in the code released in Zenodo means that I was unable to verify _any_ of the results presented in the manuscript within the time I can set aside for a manuscript review.
I regret that the authors did only the absolute minimum to ensure reproducibility of their research and did follow the lead of SciPost-Codebases in producing excellent quality manuals and code repositories accompanying their research.
In conclusion, the SciPost's requirement to "provide sufficient details (inside the bulk sections or in appendices) so that arguments and derivations can be reproduced by qualified experts" has been met in principle, but the repository provided puts needless obstacles in the reproduction process (such as insisting on not sharing the code which authors used to compute summary statistics).
Anonymous on 2025-04-28 [id 5420]
(in reply to Anonymous Comment on 2025-03-20 [id 5301])Thank you for your comments. We have updated the shared files. Now, we have shared simulation code along with the data and analysis code for H2, H4, bending energy, end-to-end distance and tangent correlations. We have also updated the readme files. It explains how to reproduce the plots from the given data and codes.We have also included the ouput files produced from the analysis code.
We hope it clarifies the concerns regarding the reproducibility of the plots.

---

## Round 2 · Referee Report · Anonymous (Referee 3) · 2025-3-19

Report
2 - It is great that the authors have introduced the flexure number that also takes into account the elasticity of the self-propelled filament. Does it mean that in the current manuscript, the flexure numbers are one or two orders of magnitude smaller than in Isele-Holder et al. (2016), i.e., that the filaments are much stiffer than in the previous work? How do the different ranges of flexure numbers affect the results?
3 - I do not entirely understand how the simulation snapshots that the authors have sent in their rebuttal letter and the one in Fig. 2(e) of the manuscript are related to the local minimum of the tangent-tangent correlation function in Fig. 6(b). I suggest that the authors extend the discussion and indicate in the simulation snapshot the arc lengths s for the local minimum and maximum in the correlation function.
Recommendation
Ask for major revision
Comment: The authors replied that the primary difference between their work and Isele-Holder et al., Soft Matter 12, 8495 (2016) is the dimensionality--and that the models are nearly identical in 2D. However, they neither cite nor discuss this work in their manuscript. Isele-Holder's publication has a very similar title and discusses the results of 3D simulations. Therefore, a reader could think that the manuscript under consideration (on 2D results) adds no new knowledge. I expect the authors to cite the study of Isele-Holder et al. in their manuscript and to discuss their findings in light of these previous results. They should state clearly where their results are in line with the 3D findings and where the different dimensionality leads to qualitatively or quantitatively different results.
Response: We have included the suggested article Soft Matter 12, 8495 (2016) in the references of the revised manuscript. We missed citing this article in the previous version, so we thank the reviewer for pointing this out. While our results share some features with those reported in Soft Matter 12, 8495 (2016), they differ in several key aspects. In three dimensions, the structural and dynamical complexity of the active polymer increases. For example, observing a beating phase in 3D is difficult without considering the filament’s torsional rigidity. Conversely, a helical phase will not appear in 2D.
In summary, the dynamics in 3D and 2D exhibit significant differences. In 2D, Soft Matter 12, 8495 (2016), authors identified distinct dynamical phases, including elongation, beating, circling, and rotation. In contrast, our 3D study reveals the presence of helical phase alongside extended and rotating states.
Beyond these differences, we have introduced a minimal model that incorporates cargo attachment as a control parameter, denoted by $\rho$. This addition underscores the novelty of our manuscript compared to previous studies. We have included a few additional lines in the Introduction to emphasize the primary distinctions. More specifically, we have stated that a 3D filament studied in the manuscript could bring different phases that are absent in the existing 2D results.
Comment: It is great that the authors have introduced the flexure number that also takes into account the elasticity of the self-propelled filament. Does it mean that in the current manuscript, the flexure numbers are one or two orders of magnitude smaller than in Isele-Holder et al. (2016), i.e., that the filaments are much stiffer than in the previous work? How do the different ranges of flexure numbers affect the results?
Response: We agree with the reviewer the calculations are performed for very stiff filaments $\l_p\ge L$. But, we have also considered the variation of the bending rigidity, that is, persistence length. By doing so, we vary the flexure number. Our study does not show any significant variation in response to changes in the flexure number. As shown in Fig. 7b, the radius of curvature as a function of $\chi$ remains consistent across different values of the filament's persistence length $l_p$. We observe the same power-law behavior with a flexure number for all persistence lengths. That concludes that changing the stiffness of the filament does not change our results, specifically in the limit of $\chi <1$.
Comment: 3 - I do not entirely understand how the simulation snapshots that the authors have sent in their rebuttal letter and the one in Fig. 2(e) of the manuscript are related to the local minimum of the tangent-tangent correlation function in Fig. 6(b). I suggest that the authors extend the discussion and indicate in the simulation snapshot the arc lengths s for the local minimum and maximum in the correlation function.
Response: In the snapshot shown in Fig. 2(e), the filament exhibits buckling at a certain distance from the tail end. This buckling results in local minima in the tangent correlation. Intuitively, this can be explained as follows: The front load opposes forward motion under compressive force, causing resistance that induces bending along the filament’s backbone. Figure 2(e) in the manuscript visually represents this buckled conformation. We have clarified this in the revised manuscript.

---

## Round 2 · Author Response

Thank you for providing the referee report on our article. We appreciate the referees' valuable comments and suggestions on our manuscript. We have carefully incorporated all the suggested revisions and provided a point-by-point response in the response file. All major corrections and responses are highlighted in blue for clarity.
We hope that the revised manuscript meets the standards for publication in SciPost Physics and look forward to your feedback.
Sincerely yours,
Prabhakar Maurya, Shalabh K Anand, and Sunil P. Singh

---

## Round 2 · List of Changes

A list of major changes. 1. A minor revision has been made to the abstract. 2. The introduction has been slightly modified, and references (31–33) have been added. 3. Equation 2 has been adjusted for clarity, along with some changes to the simulation parameters. 4. A snapshot in Fig. 2 has been added to illustrate an additional state (Fig. 2e). 5. Minor modifications have been made to Fig. 3b, Fig. 4, Fig. 7a, and Fig. 9b. Additionally, a new figure (Fig. 7b) has been included to show results as a function of the flexure number. 6. A paragraph has been added to summarize the results of Fig. 7b. 7. Typos and the definitions of H₂ and H₄ in Equation 6 have been corrected in the revised manuscript. 8. The last paragraph of the summary section has been revised to emphasize future work and the importance of hydrodynamics. 9. A link to the code and supporting movie files has been added as reference 68.

---

## Round 3 · Author Response

Dear Editor,

Thank you very much for sending us the referee report on our article. We are pleased with the comments and suggestions of the referees on our manuscript. We have incorporated all the suggestions and comments in the revised manuscript. We have provided a point-by-point response to each comment in the response file. All the major corrections and responses are highlighted in blue color.

We have provided a working code and a readme file with all the necessary details to run the code and reproduce the relevant data. The link to the repository for an updated code version is embedded in the revised manuscript.

We hope that you will find the revised manuscript suitable for publication in the journal of SciPost Physics.

We are looking forward to hearing from you.

Sincerely yours,
Prabhakar Maurya, Shalabh K. Anand, and Sunil P. Singh

---

## Round 3 · List of Changes

1. We have rephrased a paragraph (paragraph 4) in the Introduction.
  2. Figure 5-a and b is rescaled with their passive values.
  3. Paragraph 4 in the section 3.4 (Bond Correlation) is rephrased.
  4. Figure 9 is rescaled with the contour length of the filament.
  5. A new reference (62) is added.
  6. We have updated the link to the latest version of codes in the repository in the manuscript.

---

## Round 4 · Author Response

Dear Editor,
Thank you for providing the report on our article. We appreciate the comments and suggestions on our manuscript. We have carefully incorporated all the suggested revisions. All major corrections and responses are highlighted in blue for clarity.
We hope that the revised manuscript meets the standards for publication in SciPost Physics and look forward to your feedback.

Sincerely yours,
Prabhakar Maurya, Shalabh K Anand, and Sunil P. Singh

---

## Round 4 · List of Changes

1. In the fourth paragraph of the introduction, we have added a few sentences signifying the motivation behind our work.
  2. We have included one more snapshot in figure 2 (Fig. 2b).
  3. We have added a couple of sentences in the second paragraph of section 3.2. These sentences are about our observation on difference between a 2D filament pushing a load with respect to our 3D results.
  4. We have included some text comparing our 3D results with respect to the existing results on a 2D filament. The texts are in the fourth paragraph of the section named "Summary and Conclusion".
  5. We have updated the shared files. Now, we have included data files for one of the parameters ($\rho=0), as well as analysis codes. We have made a readme file which explains all the shared files and how to reproduce the results presented in the manuscript.

---

## Editorial Decision

published